# Influence of Peripheral Transluminal Angioplasty Alongside Exercise Training on Oxidative Stress and Inflammation in Patients with Peripheral Arterial Disease

**DOI:** 10.3390/jcm10245851

**Published:** 2021-12-13

**Authors:** Franziska Koppe-Schmeißer, Melanie Schwaderlapp, Julian Schmeißer, Jörn F. Dopheide, Thomas Münzel, Andreas Daiber, Christine Espinola-Klein

**Affiliations:** 1Center for Cardiology, Cardiology I—General and Interventional Cardiology and Intensive Care, University Medical Center, Johannes Gutenberg-University, 55131 Mainz, Germany; Franziska.Koppe-Schmeisser@unimedizin-mainz.de (F.K.-S.); julian.schmeisser@unimedizin-mainz.de (J.S.); tmuenzel@uni-mainz.de (T.M.); daiber@uni-mainz.de (A.D.); 2Center for Cardiology, Cardiology III—Angiology, University Medical Center, Johannes Gutenberg-University, 55131 Mainz, Germany; melanie.schwaderlapp@unimedizin-mainz.de; 3Bern University Hospital, University of Bern, 3010 Bern, Switzerland; joern.dopheide@ksgr.ch; 4Division of Angiology, Department of Internal Medicine, Cantonal Hospital Graubuenden, 7000 Chur, Switzerland; 5Vorarlberg Institute for Vascular Investigation and Treatment (VIVIT), 6800 Feldkirch, Austria; 6Department of Internal Medicine, Academic Teaching Hospital Bregenz, 6900 Bregenz, Austria

**Keywords:** reactive oxygen species, oxidative stress, peripheral arterial disease, exercise training, percutaneous transluminal angioplasty

## Abstract

In patients with intermittent claudication, exercise training ameliorates inflammation by reducing oxidative stress. A total of 41 patients with intermittent claudication (Rutherford 3) were included in the study (with 21 patients treated by endovascular revascularization (ER), and 20 patients without ER). All patients were referred to home-based exercise training. Absolute and initial claudication distance (ACD, ICD) and ABI (ankle–brachial index) were measured. ROS (reactive oxygen species) formation was measured using the luminol analogue L-012. Follow-up was performed after 3 months. ROS production after NOX2 (NAPDH oxidase 2) stimulation showed a significant reduction in both groups at follow-up (PTA group: *p* = 0.002, control group: *p* = 0.019), with a higher relative reduction in ROS in the PTA group than in the control group (*p* = 0.014). ABI measurements showed a significant increase in the PTA (peripheral transluminal angioplasty) group (*p* = 0.001), but not in the control group (*p* = 0.127). Comparing both groups at follow-up, ABI was higher in the PTA group (*p* = 0.047). Both groups showed a significant increas ACD and ICD at follow-up (PTA group: ACD: *p* = 0.001, ICD: *p* < 0.0001; control group: ACD: *p* = 0.041, ICD: *p* = 0.002). There was no significant difference between both groups at follow-up (ACD: *p* = 0.421, ICD: *p* = 0.839). Endovascular therapy in combination with exercise training leads to a lower leukocyte activation state with a reduced NOX2-derived ROS production paralleled by an improved ABI, ACD and ICD. Our data support the strategy to combine exercise training with preceding endovascular therapy.

## 1. Introduction

Inflammation is the driving force in atherosclerotic diseases [1]. Patients with peripheral artery disease (PAD) have a high frequency of other atherosclerotic manifestations in the coronary and cerebral arteries and a high inflammatory status [2,3], due the fact that this group of patients represents a generalized form of atherosclerosis. This corresponds to a high mortality rate among PAD patients [4]. Inflammation leads to the production of reactive oxygen species (ROS) by triggering a positive feedback mechanism with even further production of ROS [5], causing endothelial dysfunction, and thus aggravation of atherosclerosis [6]. Similar mechanisms were described for PAD patients [7]. Oxidative stress and inflammation markers appear to correlate with the severity of PAD [8], displaying very high levels in patients with critical limb ischemia [9]. Thus, PAD patients should be treated as soon as possible according to the severity of the disease. According to current guidelines, a conservative approach with best medical treatment and exercise is recommended for patients with intermittent claudication (Rutherford 1–3) [10]. Nevertheless, several studies highlight the efficacy of endovascular therapy on symptom relief, walking distance and quality of life in stable patients with intermittent claudication [11,12], despite the fact that this intervention may be associated with increased morbidity and mortality. Therefore, endovascular therapy should be restricted to patients who do not respond to exercise training, or when symptoms substantially alter daily life activities [10]. It is widely accepted that, in patients with intermittent claudication, exercise training is effective and improves quality of life and walking distance. We previously reported a reduced level of oxidative stress through a reduction in ROS production and inflammation markers in PAD patients under home-based exercise training [6], with distinct benefits of supervised- vs. non-supervised exercise training [13]. However, little is known about the influence of endovascular therapy, such as transluminal angioplasty (PTA), combined with exercise training on oxidative stress and inflammation in patients with PAD. Furthermore, several PAD patients were very limited at performing exercise training at the beginning of the study due to very short pain-free walking distances.

The aim of the present study is to analyze the changes in pro-inflammatory markers and level of oxidative stress in patients undergoing endovascular therapy before exercise training. We hypothesize that endovascular therapy facilitates patients with a very limited pain-free walking distance to exercise more regularly, thus leading to a reduced inflammatory status and reduced ROS production.

## 2. Materials and Methods

### 2.1. Study Population

The present study was approved by the Ethics Committee of the University of Mainz and the State of Rheinland-Pfalz, Germany (permit number: 11140). Participation was voluntary and all rules outlined in the Declaration of Helsinki were followed. All participants gave written informed consent. In total, 41 patients admitted to the Center of Cardiology of the Johannes Gutenberg University Mainz with known PAD and intermittent claudication (Rutherford Stage 3), were included in the study. Mean follow-up was 3 months. Patients with cancer, autoimmune disease or chronic inflammatory conditions were excluded from the trial. In addition, patients younger than 18 years and pregnant women were not included in the study. Hypertension was defined if patients had a previous diagnosis of hypertension (blood pressure ≥ 140/90 mm Hg), or if they receive antihypertensive treatment. Participants were classified as smokers (current or quit < 1 year), former smokers (quit ≥ 1 year) or never-smokers [14]. Patients with known and treated diabetes as well as those with a current fasting blood-glucose level >125 mg/dL were classified as having diabetes. Family history of premature atherosclerosis was defined in patients with a documented case of atherosclerotic disease (PAD, atherosclerotic stroke, coronary artery disease) in a first-degree relative before the age of 65 years. Patients with lipid-lowering therapy or with a history of cholesterol levels >240 mg/dL were defined to suffer from hyperlipoproteinemia.

For the study, we included 41 patients with intermittent claudication (Rutherford 3) with ankle–brachial index <0.9 [10]. A total of 21 patients were treated with a peripheral transluminal angioplasty (PTA), referred to as the “PTA group”. The control group consisted of 20 patients with stable intermittent claudication without endovascular revascularization (ER). The decision whether to perform a PTA was based on symptom severity in daily life and patients’ preference. All patients were referred to a home-based exercise training program [6]. Home-based exercise training was defined as a non-supervised form of exercise training. Written information on how to perform the home-based exercise training under self-management, including exercises at rest, was given to both patient groups. All patients were asked to walk for at least 30 min and up to 60 min per day, at least 3–5 days a week. They were instructed to walk with an intensity that would as close as possible reach their typical claudication sensations, then to rest for up to 5 min, and repeat the same distance at a lower intensity. This protocol is in accordance with standard exercise recommendations [15]. Patients were asked to keep a diary of their weekly training efforts. Improvements in walking distance and training efforts were interpreted from a final treadmill test compared with baseline treadmill results.

Walking distance was measured using a standard protocol, on a treadmill with 10% slope at a speed of 3.0 km/h. The pain-free walking distance (ICD = initial claudication distance) and the absolute walking distance (ACD = absolute claudication distance) of each patient were assessed at the beginning of the study and at the end of follow-up.

### 2.2. Preparation of Blood Samples

Blood samples were drawn from all patients by venipuncture after fasting for at least 12 h. Blood was either taken in Monovettes^®^ with heparin, EDTA or citrate. All samples were immediately transferred either at 0 °C or at room temperature to our research facility or the Institute for Clinical Chemistry and Laboratory Medicine, for measurement on the same day of blood sampling.

### 2.3. Measurement of Leukocyte Oxidative Burst by Chemiluminescence

The determination of ROS levels in human whole blood using a chemiluminescent dye, the luminol analogue L-012 (8-Amino-5-Chloro-7-Phenylpyrido [3,4-d] pyridazin-1,4-(2H,3H)Dion) (Wako Pure Chem. Ind., Osaka, Japan), was conducted in unstimulated and stimulated blood as previously described [16]. For induction of the oxidative burst, the blood was stimulated by phorbol 12,13-dibutyrate (PDBu) (10 μM, Sigma Aldrich). L-012 is an unspecific ROS sensor and, accordingly, reveals the totality of oxidative stress [15]. Therefore, L-012 is a suitable ROS-sensitive dye for measuring hydrogen peroxide, as a degradation product of superoxide, from oxidative burst in leukocytes, which mainly depends on the phagocytic NADPH oxidase (NOX2) [16,17]. Briefly, venous blood from citrate Monovettes^®^ was used for the assay and maintained at room temperature. The blood was diluted 1:50 in Dulbecco’s PBS (no magnesium, calcium or bicarbonate), and the activator of both protein kinase C and NOX2, PDBu, was added to the L-012 (100 μM)-containing buffer (final volume 200 µL). The basal chemiluminescence signal was determined in the absence of PDBu. The produced chemiluminescence was time-dependently determined by a chemiluminescence plate reader (Centro, Berthold Techn., Bad Wildbad, Germany) over a total time of 20 min (expressed as counts/min and normalized to the number of white blood cells). Inhibition of NOX2 activity and scavenging of superoxide by PEG-SOD were previously applied to this assay to characterize its specificity [16].

### 2.4. Clinical Chemistry Parameters

Blood sampling was performed as described above. Serum was generated using serum Monovettes^®^, immediately divided into aliquots and stored at −80 °C. The serum levels of lipids (total-, HDL- and LDL-cholesterol; triglycerides), blood glucose, HbA1c, fibrinogen and cellular counts of polymorphonuclear neutrophils (PMNs) and monocytes (total count and proportion of blood cells) were determined at the Institute for Clinical Chemistry and Laboratory Medicine on the same day of blood sampling. C-reactive protein (CRP) was measured using a highly sensitive, latex particle-enhanced immunoassay with a detection range of 0–20 mg/dL (Roche Diagnostics).

### 2.5. Statistical Analysis

For data management and statistical analysis, Microsoft Excel 2019 (Redmond, Washington, DC, USA) and SPSS statistical software 26 (IBM, Armonk, NY, USA) was used. Data are stated as “median (25th; 75th percentile)” for continuous variables in all tables, unless otherwise specified. To generate an easily interpretable graphic presentation, figures are presented as the mean with the standard deviation. A value of *p* < 0.05 was considered to be significant. Gaussian distribution of the categorical characteristics of the study population was checked by using a chi-squared test. Gaussian distribution of the other parameters was checked using the Shapiro–Wilk test.

The two groups (control vs. PTA group) were contrasted using the Mann–Whitney *U* Test. Comparison of measurements taken at admission and follow-up in each group was performed using a two-tailed paired *t*-test with a Gaussian distribution. If the normality test failed, the Wilcoxon matched-pairs non-parametric test was used instead.

## 3. Results

### 3.1. Study Population

Baseline characteristics of the study population are shown in Table 1. With regard to gender; age; cardiovascular risk factors and medication such as ACE inhibitors/sartans; anticoagulation, platelet-aggregation inhibition and statin treatment, we did not find significant differences between the PTA group in general and the control group (Table 1). In the PTA group, 12 patients underwent drug-coated balloon angioplasty of femoro-popliteal arteries. In addition, bare-metal stents were used in 5 of these 12 cases. The remaining eight patients were treated with plain old balloon angioplasty combined with bare-metal stents of iliac arteries.

In the PTA group and control group, levels of total cholesterol as well as HDL-cholesterol, LDL-cholesterol and triglycerides displayed no differences. Furthermore, HbA1c and glucose levels were similar in both groups (Table 2). In addition, similar levels of acute-phase proteins such as CRP and fibrinogen and equal levels of monocytes and polymorphonuclear monocytes (PMNs) were measured (Table 2).

### 3.2. Analysis of ROS Production

We analyzed changes in ROS production as oxidative stress is a driving force in atherogenesis. At baseline, there was no significant difference between the PTA and the control group with regard to NOX2 activity under basal conditions and PDBu-stimulated conditions. At follow-up, both groups showed a significant reduction in ROS production after NOX2 stimulation in the “basal 20 min” condition (Table 3).

Relative reduction in ROS production after NOX2 stimulation under basal conditions at follow-up was significantly higher in the PTA group than in the control group (Figure 1a). With regard to NOX2 stimulation by PDBU, both groups showed a reduction in ROS production at follow-up, but results were not statistically significant (Table 3). Concerning the relative reduction in ROS production after NOX2 stimulation with PDBU at follow-up, there was no significant difference between both groups (Figure 1b).

### 3.3. Analysis of ABI, Absolute Claudication Distance and Initial Claudication Distance

At baseline, the absolute claudication distance (ACD) and the initial claudication distance (ICD) were significantly lower in the PTA group ((a) in Table 4). As a consequence, a significantly lower ankle–brachial index (ABI) was also measured in the PTA group at baseline ((a) in Table 4).

At follow-up, we observed a significant increase in the ACD and ICD in the PTA group with exercise training (Figure 2a,b), as well as in the control group with exercise training only (Figure 2a,b). There was no significant difference with regard to ACD or ICD between the two groups at follow-up ((a) in Table 4).

With regard to ABI, there was a significant increase at follow-up in the PTA group ((b) in Table 4). In contrast, there was no significant change in ABI measurements in the control group ((c) in Table 4). Comparing ABI measurements at follow-up between the PTA and control groups, there was a significant difference in favor of the PTA group ((a) in Table 4).

## 4. Discussion

It is well known that atherosclerosis is influenced by inflammation and oxidative stress, and is therefore classified as a chronic inflammatory disease [18]. Inflammation increases the production of ROS [5,19], causing endothelial dysfunction and further aggravation of atherosclerosis [6] and resulting in a vicious circle. Therefore, it is essential to treat PAD patients as early as possible. According to international guidelines at present, exercise training is the primary recommended therapy for patients with intermittent claudication [10,20]. In our previous study, we observed a reduced level of oxidative stress by reduction in ROS production in PAD patients under home-based exercise training [6]. To our knowledge, little is known about the influence of endovascular therapy, such as PTA, combined with exercise training on oxidative stress and inflammation, in patients with PAD. Frequently, PAD patients are very limited in their ability to carry out exercise training at the beginning of treatment due to their very short pain-free walking distances. We were, therefore, interested in changes to oxidative stress and walking distance in patients undergoing endovascular therapy with PTA alongside regular exercise training.

After a mean of 3 months of training, an increased ICD and ACD compared to baseline was found in both groups. These results agreed with our previous study [6], which showed exercise training, even in a non-supervised form [13], to be successful at improving walking impairment and inflammation in PAD patients. At baseline, PAD patients, who were later referred to the PTA group, had a significantly shorter pain-free and absolute walking distance, which prevented them from performing sufficient exercise training on a regular basis. The intervention by ER was therefore necessary to enable these patients to further improve their walking impairment by carrying out their exercise program.

With regard to ABI, there was a significant increase at follow-up in the PTA group, in contrast to the control group. Comparing ABI measurements at follow-up between both groups, ABI remained significantly higher in the PTA group.

Patients in the control group started the training program with a significantly higher ABI at baseline compared with the PTA group, which agreed with the differences measured for the walking distances. The improvement in walking distance and ABI, however, was more striking for the PTA group, due to improved peripheral circulation after revascularization of major limb vessels. In contrast, the improvement in walking distance in the control group was not accompanied by an ABI improvement, due to the fact that an amelioration of their walking impairment was caused by an improved collateralization (i.e., arteriogenesis) [21], which in most cases is not observed in a regular ABI measurement. Therefore, the change observed in the follow-up period was generally stable in the control group and showed no significant change.

Concerning ROS production, both groups showed a significant reduction in basal ROS production, although the relative reduction in ROS production (% change in comparison with the value at admission) was higher in the PTA group than in the control group, which may be indicative of a worse redox state and more severe PAD score in those patients recommended for PTA. PDBu-stimulated ROS production showed a relative reduction at follow-up in both groups, but missed the point of significance.

This is consistent with our previous study, in which we observed less ROS production in PAD patients under home-based exercise training. In addition, pain-free and absolute walking distance increased [6]. A possible explanation for high ROS production in PAD patients may be based on enhanced expression of TREM-1 on PMNs of PAD patients [9], causing an intensified interaction with platelets, and thus enhancing their ROS production [7,22]. The central role of ROS formation is further supported by a previous study demonstrating that pain-free walking distance was increased, and the peak claudication pain was decreased, by therapy of 28 PAD patients with the antioxidant drug alpha-lipoic acid [23]. Finally, walking distances in 251 PAD patients showed a negative association with endothelial-cell inflammation markers but a positive association with circulating-antioxidant capacity [24].

Our data suggest that exercise training reduces the inflammatory state in PAD patients by a reduction in ROS production (or vice versa, anti-inflammatory effects of exercise prevent oxidative stress by NOX2 in leukocytes) and therefore achieves a change in absolute and pain-free walking distances. Concerning symptom relief, i.e., walking distance and quality of life, of patients with intermittent claudication, several studies showed the efficacy of endovascular therapy. Fakhry et al. [12] showed that, among patients with intermittent claudication after one year of follow-up, a combination therapy of endovascular revascularization followed by supervised exercise resulted in a significantly larger improvement in walking distances and health-related quality-of-life scores compared with supervised exercise only. A systematic review of 12 trials [11] in patients with intermittent claudication compared three treatment strategies: open surgery, endovascular therapy and exercise therapy, with medical treatment alone. All of the three alternatives were superior to medical treatment only in terms of walking distance and claudication.

Murphy et al. randomized 111 patients with intermittent claudication and lesions in the iliac aorta into groups: best medical treatment (BMT) alone, or in combination with supervised exercise training or stenting. Changes in maximal walking distance were greatest with supervised exercise training after 6 months, while stenting showed a greater improvement in peak walking time compared with BMT alone. Nevertheless, after 18 months follow-up, there was no statistically significant change in terms of peak walking time between stenting and supervised exercise training.

Therefore, choice of therapy should rely on patients’ symptoms and preferences, as well as clinical context and availability of interventional or surgical management. Endovascular therapies such as PTA should be offered to patients who are very limited in carrying out exercise training, due to a very short pain-free walking distance at the beginning of the training protocol. Our study suggests that interventional therapy with PTA should be offered to patients with intermittent claudication, who are highly symptomatic during their daily life activities due to a very limited pain-free walking distance. Interventional therapies such as PTA may lead to a longer pain-free walking distance, thus enabling patients with a more severe PAD score to perform efficient exercise training. Our study results show that PTA in combination with exercise training leads to a reduced inflammatory state by reduction in ROS production, and thus improves ABI, absolute and pain-free walking distance.

## 5. Limitations

For our study, we included patients with intermittent claudication (Rutherford 3). Patients who were very limited in their daily life activities and were symptomatic were chosen for the PTA group by clinical decision as well as the fact that ACD and ICD were significantly lower in the PTA group at baseline. Furthermore, the ABI at the beginning was significant lower in the PTA group. Therefore, there was a bias due to a clinical decision at baseline. In addition, due to the nature of home-based exercise training, the training was not performed under supervision of a specialized physiotherapist. Therefore, we could not verify how patients trained at home (lack of compliance information) and whether all training was carried out with a comparable intensity (lack of standardization). Nevertheless, we tried to limit the variance to an acceptable range by informing the patients how to train.

## 6. Conclusions

To our knowledge, we report here, for the first time, a reduced ROS production, paralleled by an improvement in clinical parameters such as ABI, ACD and ICD after a combination of endovascular therapy and exercise training. Data from our study reveal that interventional therapies such as PTA might lead to a longer pain-free walking distance in patients with intermittent claudication, thus enabling the patients to perform efficient exercise training for yet further improvements. Endovascular therapy in combination with exercise training led to a lower leukocyte activation state, as revealed by reduction in whole blood NOX2-derived ROS production, and thus improving ABI, ACD and ICD. Further studies investigating the influence of interventional therapies on inflammation and progression of PAD are needed.

## Figures and Tables

**Figure 1 jcm-10-05851-f001:**
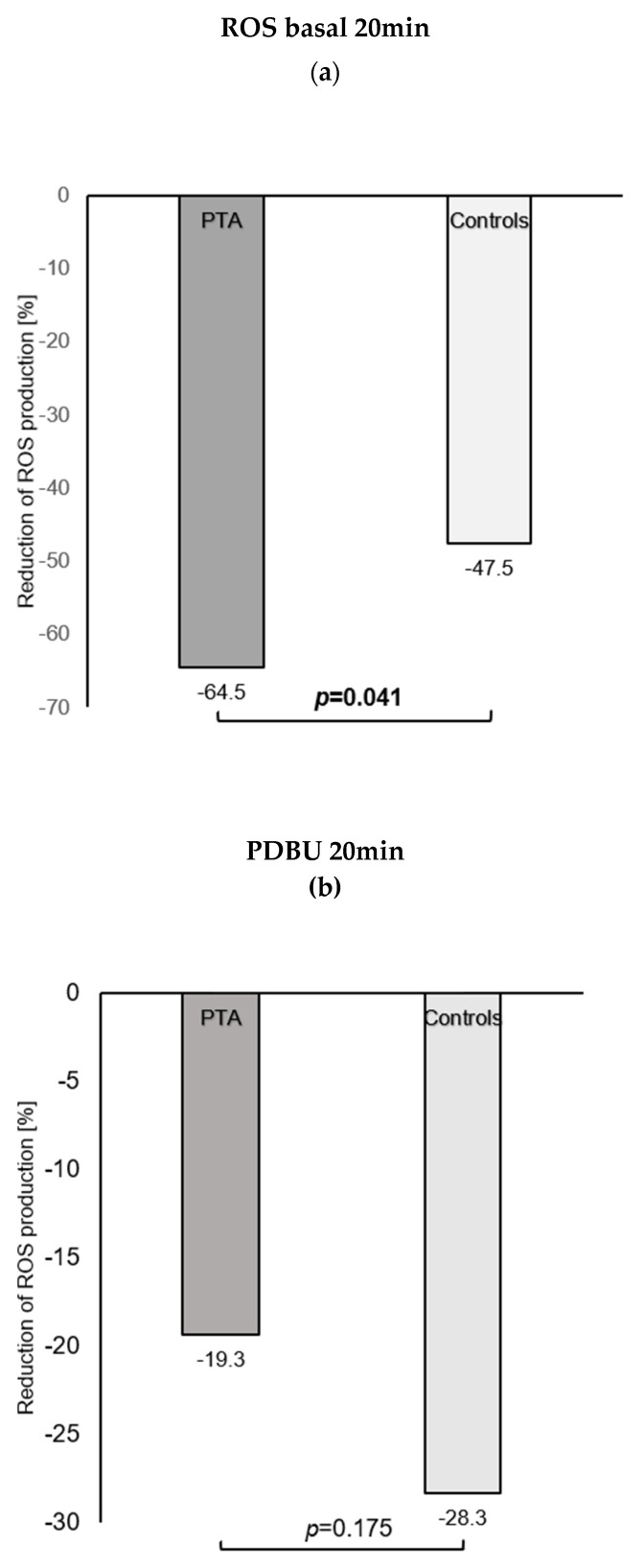
(**a**) Relative reduction in basal whole-blood (leukocyte) ROS production as % of the value at admission in the 2 treatment groups after 3 months. Data are presented as mean values. (**b**) Relative reduction in PDBu-stimulated ROS production as % of the value at admission in the 2 treatment groups after 3 months. Data are presented as mean values.

**Figure 2 jcm-10-05851-f002:**
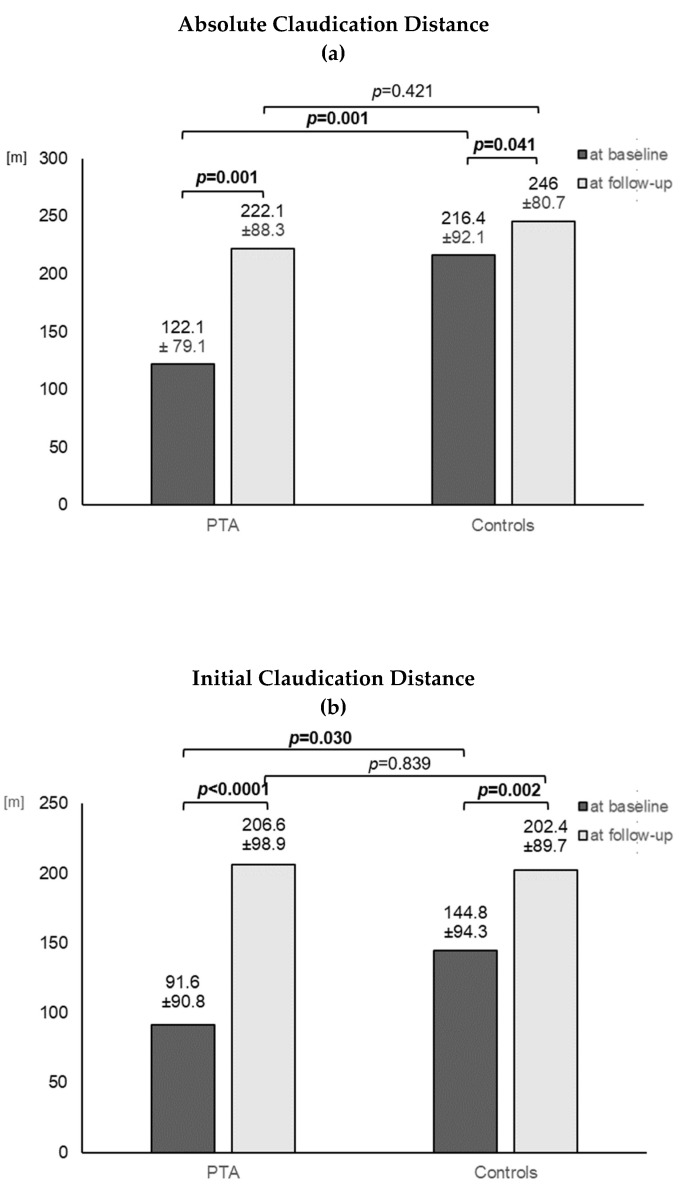
(**a**) Absolute claudication distance in the 2 treatment groups at baseline and at follow-up after 3 months with exercise training in both groups. Data are presented as mean and SD. (**b**) Initial claudication distance in the 2 treatment groups at baseline and at follow-up after 3 months with exercise training in both groups. Data are presented as mean and SD.

**Table 1 jcm-10-05851-t001:** Baseline categorical characteristics of the study population.

	Controls (*n* = 20)	PTA (*n* = 21)	*p* Value
Age (years)	68.00 (60.00; 76.00)	63.00 (58.50; 76.00)	0.51
Gender (male) (%)	13(65.00)	12 (57.14)	0.75
Hyperlipidemia (%)	9 (45.00)	8 (38.10)	0.76
Diabetes mellitus (%)	7 (35.00)	9 (42.86)	0.75
CAD (%)	6 (30.00)	4 (19.05)	0.48
Family history (%)	14 (70.00)	14 (66.67)	1.00
Active smoking (%)	8 (40.00)	10 (47.62)	0.37
Former smoking (%)	12 (60.00)	10 (47.62)	
Never smoking (%)	0	1 (4.76)	
Pack-years	47.00 (31.00; 67.25)	52.50 (37.50; 60.00)	0.71
Aspirin (%)	16 (80.00)	15 (71.43)	0.72
Clopidogrel (%)	1 (5.00)	3 (14.29)	0.61
Marcumar (%)	0	0	
NOAK (%)	1 (5.00)	3 (14.29)	0.61
ACE-inhibitors/AT1-blocker (%)	16 (80.00)	12 (57.14)	0.18
Statin treatment (%)	15 (75.00)	18 (85.71)	0.45
BMI (kg/m^2^)	27.40 (24.90; 30.30)	28.10 (25.10; 29.70)	0.74

Values are given as median (25th percentile/75th percentile).

**Table 2 jcm-10-05851-t002:** Baseline laboratory characteristics and markers of inflammation in the study population.

	Controls (*n* = 20)	PTA (*n* = 21)	*p* Value
Glucose (mg/dL)	100.00 (91.25; 124.75)	111.00 (92.50; 178.00)	0.24
HbA1c (%)	5.85 (5.63; 6.50)	6.0 0(5.70; 7.80)	0.33
Triglycerides (mg/dL)	157.50 (89.50; 275.75)	150.00 (109.00; 184.00)	0.49
Total cholesterol (mg/dL)	207.00 (147.00; 232.25)	172.00 (155.50; 197.50)	0.27
LDL cholesterol (mg/dL)	121.50 (65.25; 160.0)	93.00 (73.00; 112.50)	0.42
HDL cholesterol (mg/dL)	47.50 (40.25; 55.00)	48.00 (40.00; 62.50)	0.22
Fibrinogen (mg/dL)	347.00 (312.00; 395.75)	363.00 (310.00; 391.50)	0.38
CRP (mg/dL)	2.15 (0.93; 2.90)	1.40 (0.77; 3.85)	0.97
Leucocytes/nL	7.85 (6.11; 9.03)	7.87 (6.73; 9.38)	0.89
Monocytes (%)	6.60 (5.80; 8.00)	6.60 (5.85; 7.35)	0.82
Monocytes/mL	4.78 (4.02; 6.15) × 10^5^	5.35 (4.25; 5.78) × 10^5^	0.80
PMN (%)	61.60 (55.70; 67.40)	61.10 (53.15; 64.25)	0.56
PMN/mL	3.93 (3.20; 5.41) × 10^6^	4.27 (3.16; 5.33) × 10^6^	0.87

Values are given as median (25th percentile/75th percentile). Statistically significant changes are highlighted in bold.

**Table 3 jcm-10-05851-t003:** ROS production.

	Admission	Follow-Up	*p* Value
Controls			
Basal 20 min	66.80 (42.10; 108.60)	35.10 (19.80; 68.70)	**0.02**
PDBU 20 min	42,349.10 (28,463.20; 58,558.10)	30,313.10 (21,131.60; 46,581.10)	0.05
PTA			
Basal 20 min	57.10 (27.30; 94.10)	20.30 (15.20; 38.70)	**0.002**
PDBU 20 min	32,115.50 (23,013.70; 51,576.00)	25,915.70 (17,244.30; 34,027.10)	0.09

Values are given as median (25th percentile/75th percentile). Statistically significant changes are highlighted in bold.

**Table 4 jcm-10-05851-t004:** ABI, ACD and ICD at Baseline and after Follow-up in the PTA and Control Group.

a. ABI, ACD and ICD at Baseline and after Follow-Up.			
	Controls (*n* = 20)	PTA (*n* = 21)	*p* Value
Ankle–brachial index (ABI) (m)	0.84 (0.63;0.96)	0.58 (0.50;0.75)	**0.003**
Ankle–brachial index (ABI) FU (m)	0.73 (0.57; 0.86)	0.90 (0.75; 1.00)	**0.047**
Absolute claudication distance (m)	266.50 (123.75; 300.00)	90.00 (73.50; 135.50)	**0.001**
Absolute claudication distance FU (m)	300.00(180.00;300.00)	300.00 (138.50; 300.00)	0.42
Initial claudication distance (m)	115.00 (64.50; 235.00)	60.00 (42.50; 93.00)	**0.03**
Initial claudication distance FU (m)	200.00 (117.00; 300.00)	200.00 (105.00; 300.00)	0.84
**b. ABI, ACD and ICD at Baseline and after Follow-Up in the PTA Group.**			
	**Baseline**	**Follow-Up**	***p* Value**
Ankle–brachial index (ABI) (m)	0.58 (0.50;0.75)	0.90 (0.75; 1.00)	**0.001**
Absolute claudication distance (m)	90.00 (73.50; 135.50)	300.00 (138.50; 300.00)	**0.001**
Initial claudication distance (m)	60.00 (42.50; 93.00)	200.00 (105.00; 300.00)	**<0.0001**
**c. ABI, ACD and ICD at Baseline and after Follow-Up in the Control Group.**			
	**Baseline**	**Follow-Up**	***p* Value**
Ankle–brachial index (ABI) (m)	0.84 (0.63;0.96)	0.73 (0.57; 0.86)	0.13
Absolute claudication distance (m)	266.50 (123.75; 300.00)	300.00 (180.00;300.00)	**0.04**
Initial claudication distance (m)	115.00 (64.50; 235.00)	200.00 (117.00; 300.00)	**0.002**

Values are given as median (25th percentile/75th percentile). Statistically significant changes are highlighted in bold.

## Data Availability

The data presented in this study are available in the article. The datasets generated during and/or analyzed during the current study are available from the corresponding author on reasonable request.

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
