# Peer review of "Influence of Peripheral Transluminal Angioplasty Alongside Exercise Training on Oxidative Stress and Inflammation in Patients with Peripheral Arterial Disease"

_jcm, 2021, doi:10.3390/jcm10245851_

Round 1

Reviewer 1 Report

Dear Authors,

An interesting and novel article. The main concern is not a subject but data presentation (see below). Before consideration for further steps the following issues should be adequately addressed.

Do not begin sentences with the numbers (eg. Line 20 in 'Abstact', line 95 in the main text)

Explain all abbreviations when used the first time (ABI, ROS, PTA and NOX in 'Abstract'). They are probably obvious for the majority of the readers but please follow the common rules of writing research paper.

Why did you apply such a definition of current and past smoking. Add appropriate reference, if possible. One year after smoking cessation is more often considered to define smokers as previous ones.

Statistics. Which test was used to check data for normality. You have written that data are presented as medians but they are expressed as means with sd in Figures. Try to be consistent and correct it, please.

Results:

Avoid to duplicate data (either in text or table/figure).

Correct tables. Mind that not only changes of significance are bolded in the Tables. All are bolded.

Please write units in English. For example, 'L' instead of 'l'. Present data in the Tables more careful. There are always full stops in the decimal fractions (not commas). Decide if you want to put '0' after full stop. Once you have written and in the other places have not – be consistent.

References. Be consistent with they presentation (full or partial page number, eg. 12 vs 13).

Although, I am not a native English speaker, I found some grammar and wording mistakes. Please consult it with the native speaker familiar with medical terms.

Author Response

An interesting and novel article. The main concern is not a subject but data presentation (see below).

We thank the reviewer very much for this positive assessment of our manuscript.

Before consideration for further steps the following issues should be adequately addressed.

Do not begin sentences with the numbers (eg. Line 20 in 'Abstact', line 95 in the main text)

We changed this point and there is no sentence left which starts with a number.

Explain all abbreviations when used the first time (ABI, ROS, PTA and NOX in 'Abstract'). They are probably obvious for the majority of the readers but please follow the common rules of writing research paper.

We thank the reviewer for this note. All abbreviations are been explained now.

In detail the following sentences have been changed in the abstract line 6, 8, 12

Why did you apply such a definition of current and past smoking. Add appropriate reference, if possible. One year after smoking cessation is more often considered to define smokers as previous ones.

We agree with the reviewer that our definition of former smokers is unusual. We therefore changed the definition according to a large population based study and added this reference (http://www.ktl.fi/publications/monica/manual/index.htm). Fortunately, no patient had to be classified differently.

“Participants were classified as smokers (current or quit < 1 year), former smokers (quit ≥ 1 year), or never smokers.“

Statistics:

Which test was used to check data for normality.

Gaussian distribution of the other parameters was checked by using shapiro-wilk test

You have written that data are presented as medians but they are expressed as means with sd in Figures. Try to be consistent and correct it, please.

We thank the reviewer for this important point. To prevent misunderstanding we have rewritten the statistical explanation and hope to be clearer now.

Results:

Avoid to duplicate data (either in text or table/figure).

All duplications have been deleted.

Correct tables. Mind that not only changes of significance are bolded in the Tables. All are bolded.

Thank you for this note. We revised the tables and only significant P-values are shown in bold.

Please write units in English. For example, 'L' instead of 'l'.

We revised al units in accordance to this comment

Present data in the Tables more careful. There are always full stops in the decimal fractions (not commas). Decide if you want to put '0' after full stop. Once you have written and in the other places have not – be consistent.

We apologize for this, all tables have been checked and revised.

References.

Be consistent with their presentation (full or partial page number, eg. 12 vs 13).

All references appear now with full page numbers.

Although, I am not a native English speaker, I found some grammar and wording mistakes. Please consult it with the native speaker familiar with medical terms.

We thank the reviewer for this note. The revised manuscript has been checked by a native speaker.

Reviewer 2 Report

General comment: Important topic adressed and evaluated by the authors##

Specific comments:

Abstract:

  • In general the authors should highlight in the abstract the results they want to show with regard to the hypothesis the mentioned, so therefore it would be better the show the ROS findings before all other findings, as it is somewhat clear that hemodynamic parameters might be directly influenced by EVT and stay the same with Exercise training. The ROS results show the effect of exercise on top od EVT vs non EVT
  • ROS formation: As the authors explain already all abbreviations in the abstract, also this should be explained

Materials/Methods:

  • 21 patients underwent before the follow-up endovascular revascularisation (ER) by PTA, referred to as “PTA group”.

    What do the authors mean by PTA? What kind of EVT strategy was done (POBA only, no DCB, no Stent???). This needs some kind of explanation 

  • "the patients were referred to a home-based exercise training program"

    How was this defined? Did the patients get some detailed information about this and how was it monitored?

Results:

  • In the results section I would suggest to show the ROS results before the clinical and hemodynamic results of both groups according the comment already given with regard to the purpose of this study

Author Response

Replay to reviewer 2

General comment: Important topic addressed and evaluated by the authors

We thank the reviewer very much for this positive assessment of our manuscript.

Specific comments:

Abstract:

  • In general the authors should highlight in the abstract the results they want to show with regard to the hypothesis the mentioned, so therefore it would be better the show the ROS findings before all other findings, as it is somewhat clear that hemodynamic parameters might be directly influenced by EVT and stay the same with Exercise training. The ROS results show the effect of exercise on top od EVT vs non EVT
  • ROS formation: As the authors explain already all abbreviations in the abstract, also this should be explained

We agree with the reviewer to show fist the most important results. We changed the order of ROS and clinical results in the abstract.

We thank the reviewer for this note. All abbreviations are been explained now.

Materials/Methods:

  • 21 patients underwent before the follow-up endovascular revascularisation (ER) by PTA, referred to as “PTA group”.

What do the authors mean by PTA? What kind of EVT strategy was done (POBA only, no DCB, no Stent???). This needs some kind of explanation 

Details of revascularization have been added i the results section:

“In the PTA group 12 patients underwent drug coated balloon angioplasty of femoro-popliteal arteries. In addition, bare metal stents have been used in 5 of these 12 cases. The remaining 8 patients were treated with plan old balloon angioplasty combined with bare metal stents of iliac arteries.”

"the patients were referred to a home-based exercise training program"

How was this defined? Did the patients get some detailed information about this and how was it monitored?

We defined the home based exercise program more detailed in the methods section and added a reference.

“Home based exercise training was defined as a non-supervised form of exercise training. Written information on how to perform the home-based exercise training under self-management including exercises at rest was handed to both patient groups. All patients were asked to walk for at least 30 minutes up to 60 minutes per day, at least on 3–5 days a week. They were instructed to walk with an intensity as close as possible to reach their typical claudication sensations, then to rest for up to 5 minutes and repeat the same distance at a lower intensity. This protocol is in accordance with standard exercise recommendations [15]. Patients were asked to keep a diary of their weekly training efforts. Improvement of walking distance and training efforts were interpreted by the final tread mill test and changes of the walking distance in regard to the baseline tread mill results.”

Results:

  • In the results section I would suggest to show the ROS results before the clinical and hemodynamic results of both groups according the comment already given with regard to the purpose of this study

We agree with the reviewer and changed the order of ROS and clinical results in the results section.

Round 2

Reviewer 1 Report

I am satisfied with all corrections. The authors addressed all of them correctly. I would recommend to accept this article.

Kind regards,

B. Perek